# Does tranexamic acid lead to changes in MRI measures of brain tissue health in patients with spontaneous intracerebral haemorrhage? Protocol for a MRI substudy nested within the double-blind randomised controlled TICH-2 trial

Rob A Dineen,[1,2,3] Stefan Pszczolkowski,[1,4] Katie Flaherty,[4] Zhe K Law,[4,5] Paul S Morgan,[1,2,6] Ian Roberts,[7] David J Werring,[8] Rustam Al-Shahi Salman,[9] Tim England,[10] Philip M Bath,[4] Nikola Sprigg[4]

For numbered affiliations see end of article.

**Correspondence to**
Dr Rob A Dineen;
rob.dineen@nottingham.ac.uk

## ABSTRACT

**Objectives** To test whether administration of the antifibrinolytic drug tranexamic acid (TXA) in patients with spontaneous intracerebral haemorrhage (SICH) leads to increased prevalence of diffusion-weighted MRI-defined hyperintense ischaemic lesions (primary hypothesis) or reduced perihaematomal oedema volume, perihaematomal diffusion restriction and residual MRI-defined SICH-related tissue damage (secondary hypotheses).

**Design** MRI substudy nested within the double-blind randomised controlled Tranexamic Acid for Hyperacute Primary Intracerebral Haemorrhage (TICH)-2 trial (ISRCTN93732214).

**Setting** International multicentre hospital-based study.

**Participants** Eligible adults consented and randomised in the TICH-2 trial who were also able to undergo MRI scanning. To address the primary hypothesis, a sample size of n=280 will allow detection of a 10% relative increase in prevalence of diffusion-weighted imaging (DWI) hyperintense lesions in the TXA group with 5% significance, 80% power and 5% imaging data rejection.

**Interventions** TICH-2 MRI substudy participants will undergo MRI scanning using a standardised protocol at day ~5 and day ~90 after randomisation. Clinical assessments, randomisation to TXA or placebo and participant follow-up will be performed as per the TICH-2 trial protocol.

**Conclusion** The TICH-2 MRI substudy will test whether TXA increases the incidence of new DWI-defined ischaemic lesions or reduces perihaematomal oedema or final ICH lesion volume in the context of SICH.

**Ethics and dissemination** The TICH-2 trial obtained ethical approval from East Midlands - Nottingham 2 Research Ethics Committee (12/EM/0369) and an amendment to allow the TICH-2 MRI sub study was approved in April 2015 (amendment number SA02/15). All findings will be published in peer-reviewed journals. The primary outcome results will also be presented at a relevant scientific meeting.

## Strengths and limitations of this study

► MRI is incorporated into an international multicentre randomised controlled trial of tranexamic acid (TXA) in spontaneous intracerebral haemorrhage. This will be used to objectively assess the potential effects of TXA on cerebral ischaemia, which might not be visible using standard clinical imaging such as CT.

► This study also uses MRI to assess any potential beneficial effects of TXA on neurotoxicity and neuroinflammation.

► The inclusion of MRI necessitates the introduction of additional exclusion criteria, which increases the risk of under-recruitment and subsequent reduction of statistical power.

► A limitation of the recruitment process to the Tranexamic Acid for Hyperacute Primary Intracerebral Haemorrhage 2 MRI substudy is that recruitment took place after randomisation. As a result, survivor bias may be a potential confounder, which we address by conducting regression analyses adjusting for baseline variables.

**Trial registration number** ISRCTN93732214; Pre-results.

## INTRODUCTION AND RATIONALE

Tranexamic acid (TXA) is an antifibrinolytic drug that binds reversibly to the lysine binding site of plasminogen/plasmin,[1] leading to potent inhibition of the interaction of plasmin with fibrin that results in reduced fibrinolysis. In the context of spontaneous intracerebral haemorrhage (SICH), it has been postulated that TXA will result in more

rapid cessation of bleeding at the point of vessel rupture and hence limit haematoma expansion, a factor shown to be associated with both mortality and disability.[2] The Tranexamic Acid for Hyperacute Primary Intracerebral Haemorrhage (TICH-2) trial is an international multicentre randomised controlled trial to test the hypothesis that intravenous TXA reduces death and disability when given within 8 hours of SICH (ISRCTN93732214).[3] However, there are alternative mechanisms in addition to haemostatic by which TXA could alter the pathophysiology and hence outcome of the evolving brain injury that accompanies SICH via its interaction with the plasminogen activation axis.

First, patients with SICH are at risk of co-occurring cerebral ischaemic events and TXA could theoretically potentiate this risk by the inhibition of fibrin degradation. Diffusion-weighted imaging (DWI) MRI hyperintense lesions (DWIHLs), which are thought to be indicative of acute or subacute ischaemia, co-occur with SICH with a prevalence of 8%–35%[4–7] and are associated with a higher risk of dependence or death at 3 months (OR 4.8; 95% CI 1.7 to 13.9; P=0.004).[8] The mechanisms underlying the high prevalence of DWIHL in SICH are not known. Endothelial failure is thought to be a key event in the mechanism of small vessel disease and lacunar infarction.[9] A sequential process of endothelial failure, non-occlusive wall-associated microthrombosis, vessel rupture, microbleeding and finally reactive occlusive thrombosis leading to infarction has been proposed based on animal models.[10] The inhibition of fibrin degradation by TXA could potentiate this process by exacerbation of microvascular thrombosis. Support for a possible potentiating effect of TXA on cerebral ischaemia comes from a meta-analysis of predominantly prolonged (>10 days) TXA administration in spontaneous subarachnoid haemorrhage (SAH) which found pooled relative risk for reported cerebral ischaemia of 1.41 (95% CI 1.04 to 1.91).[11] However, multiple studies of TXA in a variety of other disease settings, including traumatic and spontaneous bleeding (non-SAH) showed no increase in cerebral ischaemia with TXA use.[12–15]

Second, there are potentially beneficial effects of TXA by modulation of the plasminogen activation cascade by reducing neurotoxicity and neuroinflammation.[16–18] A study in surgical patients has shown that TXA attenuates the inflammatory response[19] thought to be mediated by inhibition of the plasminogen activation cascade. Clinical and biological markers of inflammatory response at presentation are predictors of early neurological deterioration in SICH.[20] Perihaematomal oedema (PHO) volume and diffusion properties have been proposed as biomarkers of the inflammatory response around SICH,[21] and levels of circulating matrix metalloproteinase-3 are independently associated with PHO volume.[22] PHO increases rapidly during the first 48 hours and peaks towards the end of the second week following SICH.[23] MRI DWI of PHO most commonly shows elevated diffusion[24 25] implying that PHO results from increased permeability

of the neurovascular unit or alterations in the extravascular ultrastructural environment rather than ischaemia, although in cases where restricted diffusion is detected in PHO, an association with poor clinical outcome has been observed.[26]

In the TICH-2 MRI substudy, MRI scans will be acquired in a subgroup of the TICH-2 trial population on day 5 after randomisation (acceptable range day 2 to day 14, referred to hereafter as the day 5 scan) and day 90 after randomisation (acceptable range day 83 to day 110, referred to as the day 90 scan). The MRI data will be used to test a primary hypothesis regarding treatment-related differences in DWIHL prevalence. It will also allow secondary analyses of treatment effects on PHO volume and diffusion properties and SICH-related tissue damage at day 90.

### Primary hypothesis
Prevalence of remote DWI hyperintense lesions on the day 5 MRI scan will be greater in the TXA group compared with controls.

### Secondary hypotheses
1. Perihaematomal oedema volume and perihaematomal diffusion restriction on day 5 MRI scan will be reduced in the TXA group compared with controls.
2. SICH-related tissue damage defined on the day 90 MRI scan will be reduced in the TXA group compared with controls, controlling for initial haematoma volume.

We will test whether imaging markers of coexisting small vessel disease (cerebral microbleeds (CMBs) and white matter hyperintensities (WMHs) of presumed vascular origin) and imaging markers of cerebral amyloid angiopathy (CAA) (strictly lobar CMB, cortical superficial siderosis) are associated with the presence of DWIHL in TXA treated patients.

### DESIGN AND METHODS
### Patient population
The recruitment process is summarised in figure 1. Patients recruited to the main TICH-2 trial according to the TICH-2 trial inclusion and exclusion criteria[3] (see online supplementary information) at centres participating in the TICH-2 MRI substudy will be invited to participate in the TICH-2 MRI substudy provided they meet the additional following inclusion/exclusion criteria.
► Additional TICH-2 MRI study inclusion criteria:
   (1) Participant or delegate freely gives informed consent to participate in the TICH-2 MRI substudy or participants within the TICH-2 trial who have an MRI scan performed for clinical purposes within the MRI substudy time windows using a protocol consistent with the MRI substudy (TICH-2 trial consent includes submission of relevant clinical data which includes imaging data).

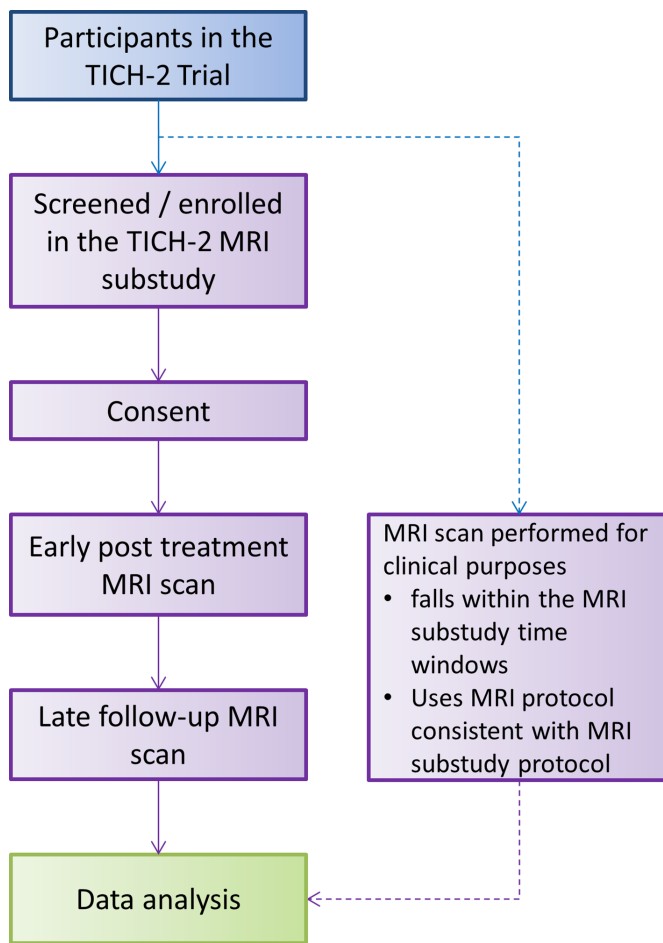

**Figure 1** TICH-2 MRI substudy flowchart. Details of participant flow for the TICH-2 trial, including the schedule for randomisation, treatment, cranial CT, clinical assessments and non-MRI follow-up assessments have been described in Sprigg et al.[3] TICH-2, Tranexamic Acid for Hyperacute Primary Intracerebral Haemorrhage 2 trial.

(2) Able to undergo MRI scanning.

► Additional TICH-2 MRI substudy exclusion criteria:
(1) Contraindication to MRI scan (eg, non-MRI-compatible implant, intraocular/intracranial metallic device or fragment, claustrophobia, etc).
(2) Clinical instability (for example cardiorespiratory or neurological instability) such that MRI scan would introduce additional clinical risk.

### Randomisation
Participants are randomised into the main TICH-2 study as described previously.[3] Recruitment into the TICH-2 MRI substudy can occur at any time between the point of initial recruitment to the main TICH-2 study (prerandomisation) up to day 7 after randomisation. Blinding to treatment allocation is maintained throughout the course of both the TICH-2 main and MRI substudy.

### Intervention
The TICH-2 trial intervention has been described previously.[3] In brief, participants are randomised (1:1) to intravenous tranexamic acid (1 g in 100 mL intravenous bolus loading dose followed by 1 g in 250 mL infusion over 8 hours) or placebo (0.9% saline using identical administration regime).

MRI scanning is performed on day 5 (acceptable range day 2 to day 14) and day 90 (acceptable range day 83 to day 110) postrandomisation. The MRI acquisition protocol (table 1) complies with the essential imaging sequences specified by the Standards for Reporting Vascular Neuroimaging standards[9] and includes two-dimensional (2D) axial, T2-weighted images, T2*-weighted images, T2-weighted fluid-attenuated inversion recovery (FLAIR) images, DWI and a three-dimensional (3D) T1-weighted volume acquisition. Axial 2D T1-weighted images are to be included if the 3D T1-weighted volume is significantly motion degraded and manufacturer optimised susceptibility weighted imaging is to be included if available.

### Outcomes
#### Primary outcome
► Prevalence of remote DWIHL on the day 5 MRI scan.

#### Secondary outcomes
► Perihaematomal oedema volume and perihaematomal diffusion restriction on the day 5 MRI scan.
► SICH-related tissue damage (the combined volume of the residual haematoma cavity and surrounding FLAIR hyperintensity) determined on the day 90 MRI scan.

Methodology for extraction of the imaging outcome measures is summarised in figure 2. DWI scans will be analysed for presence, number and distribution of DWIHL. A semiautomated method for quantification of DWIHL will be developed and validated using a subset of patients. It will be subsequently employed to identify candidate lesions based on shape and intensity features. Candidate lesions will then be accepted or rejected by at least two expert readers independently. Only DWIHL that are confirmed of low diffusion on the derived apparent diffusion coefficient maps and spatially remote from the index ICH (<20 mm) will be included as previously.[6]

A fully automated segmentation method[27] will exploit the T2, T2* and FLAIR images to segment the haematoma and PHO on day 5 MRI scan (limited for secondary hypothesis 1 to scans performed up to and including day 7), from which volume and quantification of the diffusion properties of PHO can be derived (figure 3). FLAIR images will be used to determine the volume of final haematoma cavity and surrounding hyperintensity on the day 90 scan using semiautomated segmentation supervised by experienced image analysts. T1-weighted images will be used to determine brain parenchymal volume on the day 90 scan.

WMH will be evaluated on FLAIR and T2-weighted images using an established 4-point scale[28] and automated WMH segmentation will be used for WMH

**Table 1** MRI acquisition parameters

| | DWI | T2-FLAIR | T2* GRE 2D | SWI 3D (see note 1) | 3D T1-volume | T2 | 2D T1 (see note 2) |
|---|---|---|---|---|---|---|---|
| **Orientation** | Axial | Axial | Axial | Use standard manufacturer specific SWI protocol from manufacturer's protocol tree (ie, SWAN (General Electric); VEN_BOLD (Philips); SWI (Siemens)) | Sagittal | Axial | Axial |
| **Plane** | 2D—EPI | 2D | 2D | | 3D | 2D | 2D |
| **TE (ms)** | Minimum | 125–140 | 20–30 (15–40) | | Minimum | 85–100 | 10–14 |
| **TR (ms)** | Minimum | 11 000 | 300–1000 | | Minimum | 3000–5600 | 600–650 |
| **TI (ms)** | NA | 2800 | NA | | 450– 1000 (or no inversion on old General Electric FSPGR) | NA | NA |
| **Slice thickness (mm)** | ≤4.0 | ≤4.0 | 3 (3-5) | | 1 mm isotropic voxels (1.25 mm) | ≤4.0 | ≤4.0 |
| **Slice gap (mm)** | ≤0.4 | ≤0.4 | ≤0.3 (0–1) | | NA | ≤0.4 | ≤0.4 |
| **Acquisition matrix (RFOV acceptable)** | 96–128 | 180–256 (180-512) | 180–256 | | 128–256 | 180–256 (180-512) | 180–256 |
| **FOV (mm)** | 230–240 | 230–240 | 230–240 | | 224–256 | 230–240 | 230–240 |
| **Flip angle (excitation)** | 90° | 90° | 15–60° | | 8–15° | 90° | 90° |
| **Flip angle (refocusing)** | 180° | 180° | NA | | NA | 180° | 180° |
| **b-value** | 1000 | NA | NA | | NA | NA | NA |
| **Number of slices** | 32–42 | 32–42 | 32–42 | | 180–192 (128–256) | 32–42 | 32–42 |

Preferred parameters are listed, with acceptable range of values given in brackets for sites where the preferred parameters cannot be achieved for technical reasons .

Note 1: If available.

Note 2: Only required if 3D T1-volume acquisition is degraded by patient motion.

DWI, diffusion-weighted image; EPI, echo-planar imaging; FLAIR, fluid-attenuated inversion recovery; FOV, field of view; FSPGR, fast spoiled gradient-echo; GRE 2D, gradient echo two dimensional; NA, not applicable; RFOV, rectangular field of view; SWI 3D, susceptibility weighted imaging three dimensional; TE, echo time; TI, inversion time; TR, repetition time.

volume calculation and creation of a distribution map of WMH. CMB number and distribution will be rated using Microbleed Anatomical Rating Scale and used for classification of ICH patients into probable CAA and non-CAA groups.[29]

## Sample size estimates

### Primary hypothesis

Based on previous studies finding prevalence of DWIHL of 20% in SICH, we performed calculations for sample size to allow us to detect different percentage increases in DWIHL prevalence in the TXA group above a 20% baseline, using the sample size formula for prevalence:

$$n = \frac{\left(z^2_{1-\frac{\alpha}{2}}\right) \cdot p(1-p)}{m^2}, \qquad (1)$$

where $z$=CI, P=prevalence and m=margin of error (standard is 5%). Assuming a 10% relative increase in prevalence of DWIHL (ie, from 20% to 22%) in the TXA

group, with 5% significance and 80% power, a sample size of 264 is required. Allowing 5% imaging data rejection (eg, due to excessive patient motion), a sample size of 280 is required.

### Secondary hypotheses

Our pilot data from the TICH-1 study[30] showed that for every 1 mL of SICH volume, the mean relative PHO volume increased between the day 0 and day 2 CT scan by 0.29 mL (SD, SD 0.34) in TXA-treated patients and by 0.41 mL (SD 0.27 mL) in the placebo group. Assuming a similar effect size is present on day 5 MRI scan using the sample size proposed for the main hypothesis (n=280), we will be able to detect a group difference in the increase in relative PHO with power of 0.89 and α=0.05 (independent samples t-test, two tailed).

## Statistical analyses

Group baseline characteristics will be compared between the TICH-2 MRI substudy and the main TICH-2 trial to

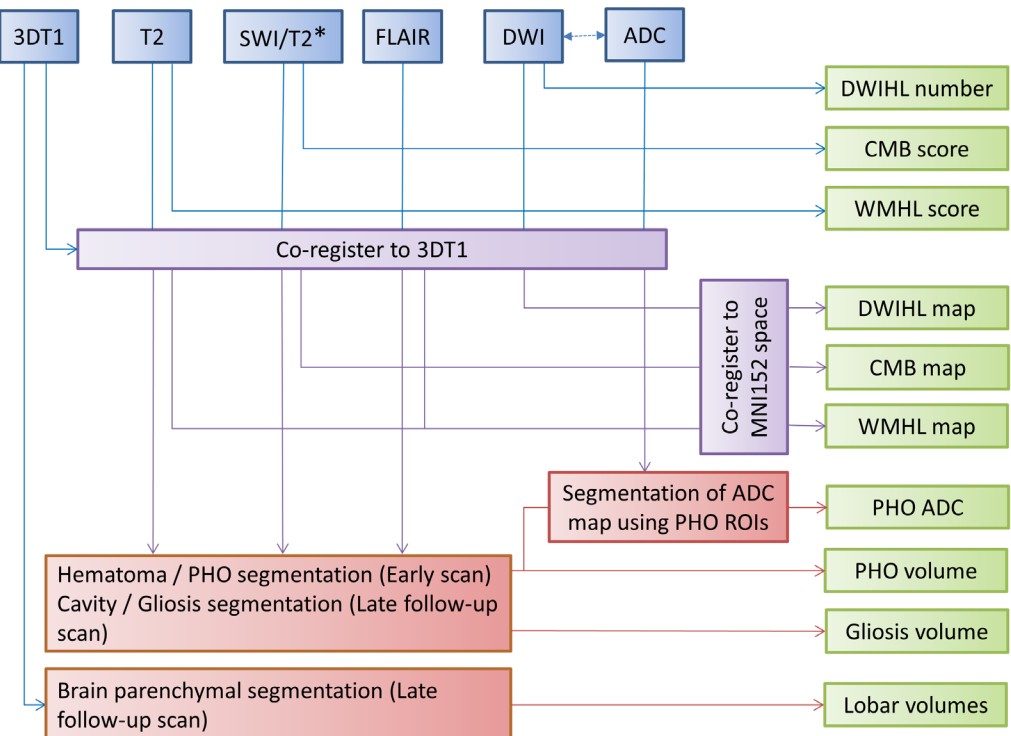

**Figure 2** Summary of image processing and outputs in the TICH-2 MRI substudy. 3D, three dimensional; ADC, apparent diffusion coefficient; CMB, cerebral microbleed; DWI, diffusion-weighted image; DWIHL, diffusion-weighted image hyperintense lesion; FLAIR, fluid-attenuated inversion recovery; MNI, montreal neurological institute; PHO, perihaematomal oedema; ROI, region of interest; SWI, susceptibility weighted imaging; TICH-2, Tranexamic Acid for Hyperacute Primary Intracerebral Haemorrhage 2 trial; WMHL, white matter hyperintense lesion.

examine the extent to which the TICH-2 MRI substudy participants are representative of the TICH-2 trial populations. To analyse the primary hypothesis, presence of DWIHL post-SICH will be compared between treatment groups using a binary logistic regression with adjustment for a selection of baseline covariates; age, time from onset to randomisation, stroke severity (using National Institutes of Health Stroke Scale), mean systolic blood pressure, known history of antiplatelet treatment and baseline haematoma volume.

To test the secondary hypotheses, a group comparison test of PHO volume, PHO diffusion metrics and day 90

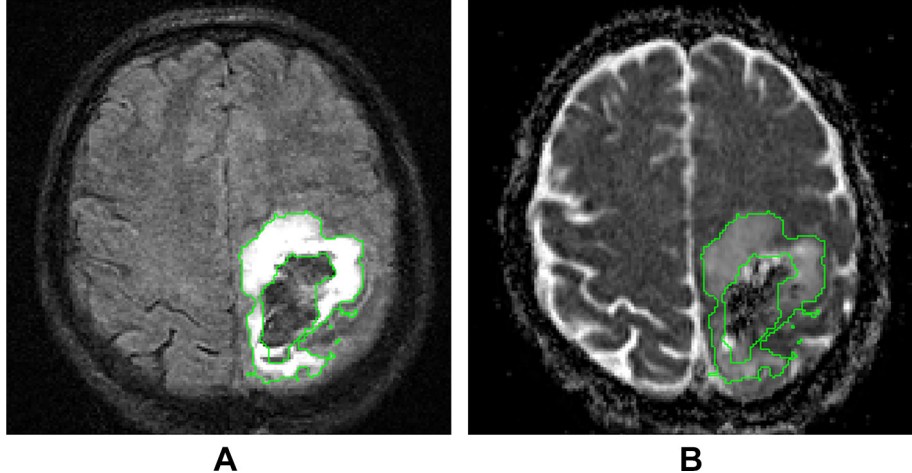

**Figure 3** Measurement of PHO volume and ADC. (A) Axial FLAIR image in native T1 space showing the PHO delineation from which the PHO volume can be derived. (B) ADC map in native T1 space. PHO ADC values are calculated from the voxels within the PHO delineation in the ADC map. ADC, apparent diffusion coefficient; FLAIR, fluid-attenuated inversion recovery; PHO, perihaematomal oedema.

combined haematoma cavity and surrounding FLAIR hyperintensity volume between participants in the TXA and control treatment groups will be performed. Other proposed analyses will include analyses of association between presence of DWIHL and imaging markers of small vessel disease (CMB, WMH) and likely diagnostic classification for CAA (based on CMB distribution and presence of superficial siderosis) and exploratory analyses of associations between imaging outcomes and clinical outcomes, with adjustment for haematoma expansion. All regression analyses will be adjusted for the covariates listed previously.

### Study organisation and funding

The TICH-2 MRI substudy is funded by a grant from British Heart Foundation (grant reference PG/14/96/31262). The TICH-2 trial is funded by a grant from the UK National Institute for Health Research Health Technology Assessment programme (project code 11_129_109). The TICH-2 trial (ISRCTN50867461) obtained ethical approval from Nottingham-2 Research Ethics Committee on 19/11/12 and an amendment to allow the TICH-2 MRI substudy was approved on 21/04/2015 (amendment number SA02/15). The University of Nottingham acts as the trial sponsor. The trial steering committee for the TICH-2 trial oversees the TICH-2 MRI substudy. Data monitoring for the TICH-2 trial is performed by an independent data monitoring committee.[3] Recruitment into the MRI substudy started in July 2015 and finished in September 2017.

### DISCUSSION

Tranexamic acid may have biological effects in the brain tissue that could alter the outcome in SICH patients, independent to the postulated effects on haematoma expansion. The main theoretical concern is that the haemostatic effects of TXA might increase the risk of cerebral ischaemia. Understanding these effects is vital, particularly if the results of ongoing trials[31] show an effect of TXA on outcome that is not (or only partially) explained by an effect of TXA on haematoma expansion. This study allows us to quantify treatment-related brain changes (particularly prevalence of ischaemic lesions, PHO volume and diffusion characteristics and late post-SICH tissue damage) in SICH patients in the context of a multicentre randomised controlled trial. If the results of the ongoing trials support the use of TXA, then the opportunity to study the potential therapeutic effects of TXA in acute cerebrovascular brain injury in a controlled trial is unlikely to occur again. By including imaging markers of small vessel disease and CAA in the analysis of DWIHL burden, we will be able to test whether there is a subgroup of patients at risk of DWIHL following TXA treatment for SICH, which will inform future studies of TXA in SICH and could ultimately influence clinical practice. If the study shows a beneficial effect of TXA on PHO, then this study may trigger studies of TXA in other acute brain pathologies associated with oedema and inflammation.

A limitation of the recruitment process to the TICH-2 MRI substudy is that recruitment took place after randomisation. As a result, survivor bias could be a confounder as a treatment allocation-related impact on survival of one group over the other could lead to imbalance between the groups entering into the study. Recruitment to the MRI substudy prerandomisation was discussed by the investigators during the study design but was considered to be impractical due to the requirement for additional information to be delivered to the participant/relatives and additional consent prerandomisation which could potentially lead to delays to randomisation for the main TICH-2 trial. To identify a potential survivor bias, we will compare group baseline characteristics between the participants in the TICH-2 MRI substudy and the main TICH-2 trial. To attempt to control for possible imbalances, we conduct the regression analyses adjusting for prespecified baseline variables, as detailed in the 'statistical analyses' section.

### SUMMARY AND CONCLUSION

TXA is a widely available and inexpensive antifibrinolytic drug, which has the potential of reducing neurotoxicity and neuroinflammation in SICH. However, it is also possible that it may potentiate the risk of ischaemic events. The TICH-2 MRI substudy will assess the effectiveness of TXA in reducing inflammatory response, while also determining whether there is an increased risk of ischaemic events following its administration. This will inform future studies of TXA and, if found to be effective and safe, have an important impact in clinical practice.

**Author affiliations**
[1]Radiological Sciences, Division of Clinical Neuroscience, University of Nottingham, Nottingham, UK
[2]Sir Peter Mansfield Imaging Centre, University of Nottingham, Nottingham, UK
[3]NIHR Nottingham Biomedical Research Centre, Nottingham, UK
[4]Stroke Trials Unit, Division of Clinical Neuroscience, University of Nottingham, Nottingham, UK
[5]Department of Medicine, National University of Malaysia, Kuala Lumpur, Malaysia
[6]Medical Physics and Clinical Engineering, Nottingham University Hospitals NHS Trust, Nottingham, UK
[7]Clinical Trials Unit, London School of Hygiene and Tropical Medicine, London, UK
[8]Stroke Research Centre, University College London, London, UK
[9]Centre for Clinical Brain Sciences, University of Edinburgh, Edinburgh, UK
[10]Vascular Medicine, Division of Medical Sciences and GEM, University of Nottingham, Nottingham, UK

**Contributors** RAD, KF, PSM, IR, DJW, RAS, TE, PMB and NS designed the study and were coapplicants for funding. SP designed and performed the image analyses. KF designed and performed the statistical analyses. RAD, SP, ZKL, KF and NS drafted the manuscript. All authors reviewed and commented on the manuscript.

**Funding** This work was supported by the British Heart Foundation [grant number PG/14/96/31262] and Health Technology Assessment Programme [grant number 11_129_109].

**Competing interests** None declared.

**Patient consent** Not required.

**Ethics approval** Nottingham-2 Research Ethics Committee.

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
