## [Reviewer comments · BMJ Open]

ARTICLE DETAILS

TITLE (PROVISIONAL)	Does tranexamic acid lead to changes in MRI-measures of brain tissue health in patients with spontaneous intracerebral haemorrhage? Protocol for a MRI sub-study nested within the double-blind randomised controlled TICH-2 trial
AUTHORS	Dineen, Rob; Pszczolkowski, Stefan; Flaherty, Katie; Law, Zhe; Morgan, Paul; Roberts, Ian; Werring, David; Salman, Rustam; England, Tim; Bath, Philip; Sprigg, Nicola

VERSION 1 – REVIEW

REVIEWER	Xia Wang The George Institute for Global Health
REVIEW RETURNED	22-Oct-2017

GENERAL COMMENTS	The authors aim to investigate whether the antifibrinolytic drug tranexamic acid (TXA) is effective to increase hyperintense ischaemic lesions in patients with spontaneous intracerebral haemorrhage (SICH). The rationale is plausible and the paper is very well written. However, there are a few things need to be clarified. 1. Please clarify how to recruit patients in the substudy to make it representative of TICH-2 trial.2. The sample size calculation statement is confusing. Does it mean the following parameters: P in TXA group is 30%, P in placebo group is 20%, 80% power, 5% significance and 5% drop-out rate? If my understanding is correct, the sample size requires much more than 280.
--

REVIEWER	Atsushi Shiraishi Kameda Medical Center, Kamogawa, Japan
REVIEW RETURNED	02-Nov-2017

GENERAL COMMENTS	This reviewer was grateful to the opportunity to review this study protocol article and considered that the study was carefully designed and the article was well-written. However, I had several concerns and suggestions to improve the quality of article. Major point Randomization of the original TICH-2 study will not be maintained sufficiently in this nested substudy. The reason why the randomization breaks is chance of outcome measurement (MRI) may be depended on assigned intervention (TXA or control), if significant intergroup difference in mortality or dependencies exists.
---

	In other words, patients in TXA group have greater chance of being scanned by MRI if TXA improved clinical outcome, or vice versa. It is so-called "survivor's bias". I believed that authors recognized this concern partially because they planned regression analysis for assessment of primary outcome, however it was insufficient to adjust for survivor's bias. Furthermore, planned analyses for secondary outcome seems to include only "univariable" analysis which I consider to be inadequate. Moreover, I considered that patients who experience neurological worsening and can not be scanned by MRI may have greater risk of DWIHL. Inevitable exclusion of these population can degrades detectability of primary outcome and increases required sample size. I recommended authors to consider the survivor's bias and to include appropriate sensitivity analyses.
--	--

REVIEWER	David Faraoni, MD, PhD, FCCP, FAHA Department of Anesthesia and Pain Medicine Hospital for Sick Children University of Toronto Toronto, Canada
REVIEW RETURNED	02-Nov-2017

GENERAL COMMENTS	In this manuscript, the authors described a study protocol that will assess the potential effect of TXA on the incidence of new ischemic lesions or reduces perihaematomal oedema in the context of SICH. The main study protocol was published in 2016 (Int J Stroke), while this manuscript describes the MRI part of the study The risk of thrombosis associated with TXA is very hypothetical, but the current analysis will allow the authors to objectively address an extensively debated topic. The major limitation of the study is the timing of the MRI. The first MRI will be performed 5 days after enrolment, while the study drug will only be administered on day 0. In the meantime, several factors could explain the potential changes the authors will observe on the MRI and it could be challenging to conclude that the only difference observed is explained by the treatment drug. Page 4 - Line 9: Not sure the authors will be looking at 'prevalence'. I would suggest to use 'incidence' instead. Page 6 - Line 35: 'Firstly, there is potential for TXA to precipitate ischaemic events'. There is no evidence (that I'm aware of) to support that sentence. Please, add a reference. Page 9: Would be useful to add a paragraph summarizing the study protocol, and the randomization part, as not all readers will have access to the Int J Stroke article.
--

VERSION 1 – AUTHOR RESPONSE

Reviewer: 1

The authors aim to investigate whether the antifibrinolytic drug tranexamic acid (TXA) is effective to increase hyperintense ischaemic lesions in patients with spontaneous intracerebral haemorrhage

(SICH). The rationale is plausible and the paper is very well written. However, there are a few things need to be clarified.

1. Please clarify how to recruit patients in the substudy to make it representative of TICH-2 trial.

Response: Please see the response to the major point relating to survivor bias raised by reviewer 2 below, which covers this point.

2. The sample size calculation statement is confusing. Does it mean the following parameters: P in TXA group is 30%, P in placebo group is 20%, 80% power, 5% significance and 5% drop-out rate? If my understanding is correct, the sample size requires much more than 280.

Response: Our apologies for the confusion: we mean a relative increase of prevalence by 10% on the assumed placebo group rate of 20%, meaning a prevalence rate of 22% rather than 30% assumed by the reviewer. This accounts for the discrepancy in the reviewer's calculation. To clarify this we have edited the sentence in the sample size estimates section to read:

'Assuming a 10% relative increase in prevalence of DWIHL (i.e. from 20% to 22%) in the TXA group...'

Reviewer: 2

This reviewer was grateful to the opportunity to review this study protocol article and considered that the study was carefully designed and the article was well-written. However, I had several concerns and suggestions to improve the quality of article.

Major point

Randomization of the original TICH-2 study will not be maintained sufficiently in this nested substudy. The reason why the randomization breaks is chance of outcome measurement (MRI) may be depended on assigned intervention (TXA or control), if significant intergroup difference in mortality or dependencies exists. In other words, patients in TXA group have greater chance of being scanned by MRI if TXA improved clinical outcome, or vice versa. It is so-called "survivor's bias". I believed that authors recognized this concern partially because they planned regression analysis for assessment of primary outcome, however it was insufficient to adjust for survivor's bias. Furthermore, planned analyses for secondary outcome seems to include only "univariable" analysis which I consider to be inadequate. Moreover, I considered that patients who experience neurological worsening and cannot be scanned by MRI may have greater risk of DWIHL. Inevitable exclusion of these population can degrades detectability of primary outcome and increases required sample size. I recommended authors to consider the survivor's bias and to include appropriate sensitivity analyses.

Response: The reviewer raises an important issue. We acknowledge that survivor bias is a potential limitation to the study and have now added a bullet point to this effect in the 'Strengths and Limitations section:

'A limitation of the recruitment process to the TICH-2 MRI sub-study is that recruitment can take place after-randomisation. As a result survivor bias may be an issue which we address by conducting regression analyses adjusting for baseline variables'

We have added the following sentence at the beginning of the Statistical Analyses section:

'Group baseline characteristics will be compared between the TICH-2 MRI sub-study and TICH-2 trial to test the extent to which the TICH-2 MRI sub-study participants are representative of the TICH-2 Trial populations.'

We have also added the following sentences to the Discussion section:

'A limitation of the recruitment process to the TICH-2 MRI sub-study is that recruitment can take place after-randomisation. As a result survivor bias may be an issue as a treatment allocation-related impact on survival of one group over the other could lead to imbalance between the groups entering into the study. Recruitment to the MRI sub-study pre-randomisation was discussed by the investigators during the study design but was considered to be impractical due to the requirement for additional information to be delivered to the participant / relatives and additional consent pre-randomisation which could potentially lead to delays to randomisation for the main TICH-2 trial. To identify a potential survivor bias we will compare group baseline characteristics between the participants in the TICH-2 MRI sub-study and the main TICH-2 trial, and to attempt to control for this we conduct the regression analyses adjusting for baseline variables.'

We have removed the work 'univariate' from the second paragraph in the statistical analysis section. This was a typing error from an earlier version of the manuscript draft, and we agree with the reviewer is not relevant for this analysis.

Reviewer: 3

Comment: In this manuscript, the authors described a study protocol that will assess the potential effect of TXA on the incidence of new ischemic lesions or reduces perihematomal oedema in the context of ICH. The main study protocol was published in 2016 (Int J Stroke), while this manuscript describes the MRI part of the study. The risk of thrombosis associated with TXA is very hypothetical, but the current analysis will allow the authors to objectively address an extensively debated topic.

The major limitation of the study is the timing of the MRI. The first MRI will be performed 5 days after enrolment, while the study drug will only be administered on day 0. In the meantime, several factors could explain the potential changes the authors will observe on the MRI and it could be challenging to conclude that the only difference observed is explained by the treatment drug.

Response: The reviewer raises an important point. The timing of the first MRI was discussed at length by the co-investigators at the time of study design. Although it is reasonable to assume that ischaemic lesions developing in ICH patients following TXA administration (if indeed these do occur) appear early after the drug administration, the true timeframe over which they develop is not known.

Performing MRI scans very early after ICH can be challenging as the participants may not be stable enough to safely undergo the procedure.

Furthermore, for our secondary hypothesis relating to perihematomal oedema, we are aware that the oedema evolves rapidly over the first few days, and hence imaging too early may impact on our study of perihematomal oedema characteristics. The choice of day 5 for the first MRI scan is therefore a compromise, but a carefully considered one that should allow us report and compare the prevalence of DWI hyperintense lesions in the TXA and placebo groups towards the end of the acute post-ICH period, as well as studying the oedema characteristics.

Comment: Page 4 - Line 9: Not sure the authors will be looking at 'prevalence'. I would suggest to use 'incidence' instead.

Response: We have considered this point carefully, and feel that what we are measuring is prevalence of DWI hyperintense lesions, as we are simply reporting the frequency at a single time

point, rather than incidence, for which we would need a MRI scan before the intervention to allow us to calculate the rate of newly occurring lesions.

Comment: Page 6 - Line 35: 'Firstly, there is potential for TXA to precipitate ischaemic events'. There is no evidence (that I'm aware of) to support that sentence. Please, add a reference.

Response: We have revised this sentence to:

'Firstly, patients with SICH are at risk of co-occurring cerebral ischaemic events, and the inhibition of fibrin degradation by TXA theoretically could potentiate this risk.'

We have used the word 'theoretically' to acknowledge the fact that there is no firm evidence on which to base the statement (and hence cannot add a reference), but in the rest of this paragraph we discuss the co-occurrence of acute cerebral ischaemic events and cite the Cochrane review that identified an elevated relative risk of 1.41 (95% CI 1.04-1.91) of cerebral ischaemic in subarachnoid haemorrhage patients treated with TXA [Baharoglu MI, et al, 2013].

Page 9: Would be useful to add a paragraph summarizing the study protocol, and the randomization part, as not all readers will have access to the Int J Stroke article.

Response: To address this point we have created a short supplementary information document briefly detailing the design of the TICH-2 trial and listing inclusion and exclusion criteria, based on the published protocol. We felt that it would better to provide this information as a supplementary file rather than burdening the main text with detail published previously. However, if the editors feel that this information is best incorporated in the main text, then we are happy to add this accordingly

VERSION 2 – REVIEW

REVIEWER	Xia Wang The George Institute for Global Health
REVIEW RETURNED	28-Nov-2017
GENERAL COMMENTS	none
REVIEWER	Atsushi Shiraishi Kameda Medical Center, Japan
REVIEW RETURNED	12-Dec-2017
GENERAL COMMENTS	I appreciated to the authors for adequate responses to my concern, a survivor bias. I considered that an addition of statement for the study limitations and planning of sensitivity analysis is appropriate. In addition, this reviewer recommends authors to describe detailed reasons why patients are excluded from MRI scan in the participant selection tree figure.
REVIEWER	David Faraoni Hospital for Sick Children, Toronto, Canada
REVIEW RETURNED	06-Dec-2017
GENERAL COMMENTS	The authors adequately answered the reviewers' comments.